# Accessibility and quality of drug company disclosures of payments to healthcare professionals and organisations in 37 countries: a European policy review

Piotr Ozieranski  ,[1] Luc Martinon,[2] Pierre-Alain Jachiet,[2] Shai Mulinari[3]

¹Department of Social and Policy Sciences, Centre for the Analysis of Social Policy, University of Bath, Bath, UK
²Euros for Docs, Paris, France
³Department of Sociology, Lunds Universitet, Lund, Sweden

**Correspondence to**
Dr Piotr Ozieranski;
p.ozieranski@bath.ac.uk

## ABSTRACT

**Objectives** To examine the accessibility and quality of drug company payment data in Europe.

**Design** Comparative policy review of payment data in countries with different regulatory approaches to disclosure.

**Setting** 37 European countries.

**Participants** European Federation of Pharmaceutical Industries and Associations, its trade group and their drug company members; eurosfordocs.eu, an independent database integrating payments disclosed by companies and trade groups; regulatory bodies overseeing payment disclosure.

**Main outcome measures** Regulatory approaches to disclosure (self-regulation, public regulation, combination of the two); data accessibility (format, structure, searchability, customisable summary statistics, downloadability) and quality (spectrum of disclosed characteristics, payment aggregation, inclusion of taxes, recipient or donor identifiers).

**Results** Of 30 countries with self-regulation, five had centralised databases, with Disclosure UK displaying the highest accessibility and quality. In 23 of the remaining countries with self-regulation and available data, disclosures were published in the portable document format (PDF) on individual company websites, preventing the public from understanding payment patterns. Eurosfordocs.eu had greater accessibility than any industry-run database, but the match between the value of payments integrated in eurosfordocs.eu and summarised separately by industry in seven countries ranged between 56% and 100% depending on country. Eurosfordocs.eu shared quality shortcomings with the underlying industry data, including ambiguities in identifying payments and their recipients. Public regulation was found in 15 countries, used either alone (3), in combination (4) or in parallel with (8) self-regulation. Of these countries, 13 established centralised databases with widely ranging accessibility and quality, and sharing some shortcomings with the industry-run databases. The French database, Transparence Santé, had the highest accessibility and quality, exceeding that of Disclosure UK.

**Conclusions** The accessibility and quality of payment data disclosed in European countries are typically low, hindering investigation of financial conflicts of interest.

### Strengths and limitations of this study

► We investigate the quality and accessibility of drug company payment disclosure data in 37 European countries.

► We use a set of measures relevant for countries with industry self-regulation, public regulation and a combination of the two.

► We present our results as a 'heat map', showing the least and most problematic aspects of payment data accessibility and quality.

► One key limitation is that that we did not quantify some aspects of the accessibility and quality of payment data.

Some improvements are straightforward but reaching the standards characterising the widely researched US Open Payments database requires major regulatory change.

## INTRODUCTION

Financial conflicts of interest (FCOIs) can bias healthcare research, practice, education and policy.[1–3] The last decade has seen a global trend towards addressing concerns about FCOIs by publishing drug company payments to the healthcare sector.[4–8] It is best exemplified by the US Sunshine Act establishing Open Payments, a database triggering extensive research on payment distribution,[9 10] and its links with drug prescription[11] and cost.[12 13] Open Payments increases transparency of FCOIs by enabling cross-checking information collected by professional organisations,[14] conference organisers[15] and scientific journals.[16] It also aids identifying corruption by highlighting unusual payment patterns.[17 18]

Unlike the USA, in most European countries, drug company payments are disclosed via industry self-regulation.[4 6] In Europe, the prevalent form of self-regulation draws on

the Code of Practice of the European Federation of Pharmaceutical Industries and Associations (EFPIA), with its minimum requirements transposed into the codes of EFPIA's national trade group members.[19] Self-regulation allows the industry to develop, implement, and oversee the rules of payment disclosure.[4 20] Compared with the US Sunshine Act, one key shortcoming of self-regulation, resulting from the industry's interpretation of European privacy laws, is making company disclosures conditional on consent granted by payment recipients.[21–23] Other problems include broader, and therefore difficult to interpret, payment categories (grants and donations, contributions to costs of events, fees for service and consultancy),[22] which are also fewer than in the USA, excluding royalties, ownership and investments. Additionally, research payments are only disclosed as lump sums per company without named recipients.[5 24] One advantage of self-regulation is a greater scope of covered healthcare professionals, including not only physicians but also nurses (to be included in the USA starting from 2022),[25] pharmacists and others.[5 21] Furthermore, self-regulation includes, like in the USA, not only hospital recipients of payments but also general practice surgeries, professional associations and other healthcare organisations.[5 24]

Only few European countries, including France, Portugal and Latvia, use government regulation, principally legislation, to impose disclosure requirements for donors and recipients, including mandatory disclosure.[4 6] Finally, one country, the Netherlands, has been identified as using a combination of self and public regulation, with the disclosure regulations developed with government's input, but lacking a legal basis and enforced via self-regulation.[4]

The scrutiny of European payment data has been limited, except for case studies of payment distribution in the UK,[21 24] Germany[26] and Ireland,[27] and a comparative analysis of payments shares not disclosed by recipients in the UK, Germany, Sweden, Switzerland, Italy, Ireland and Spain.[28] However, France is the sole country where relationships between payments and prescribing have been investigated.[29] Similarly, the potential for detecting organisational-level FCOIs is unrealised, with only two studies examining discrepancies in payments reported separately by companies and some healthcare providers[30] and commissioners[31] in England. Furthermore, corrupt relationships identified via official investigations pertaining to Greece,[32] Poland and Russia[33] might have been revealed earlier by examining payment patterns, following the US' example.[17 18] Therefore, the evidence base for any policy reform is thin, leaving the industry as the only stakeholder likely to have in-depth understanding of payment data, particularly in countries with self-regulation.

The likely reasons behind the scant disclosure research are the low accessibility and quality of payment data. Regarding accessibility, a study of European disclosure approaches has found that of six countries with self-regulation five lacked centralised payment databases.[4] In one of these countries,

Germany, the dispersal of disclosures on drug company websites was a major obstacle in data analysis.[19 26 28] A recent remedial initiative by activist data scientists has involved creating a database called eurosfordocs.eu. Inspired by a similar German project,[26] eurosfordocs.eu integrates data disclosed separately by many companies in several countries with self-regulation.[28 34] Contrastingly, of the four countries identified as having government regulation or combining it with self-regulation three had databases integrating payments reported by all companies.[4]

A related aspect of low accessibility both in countries with self-regulation and government regulation is poor user interface.[4] Of the six studied countries with self-regulation only Disclosure UK, the database run by the Association of the British Pharmaceutical Industry (ABPI), was judged as user friendly.[4] However, of the three databases in countries using government regulation or combining it with self-regulation the Dutch and Portuguese databases were described as 'partially' user friendly, while the French was deemed 'not' user friendly.[4] Challenges in the interface of the French database were only addressed by the independent data platform eurosfordocs.fr, stimulating journalistic investigations into FCOIs.[35–37]

The second problem, payment data quality, has only been examined in countries with self-regulation. For example, analyses of Disclosure UK revealed inconsistencies in reporting of payment values and recipients,[21 24] compounded by the absence of unique recipient identifiers.[38] Similar shortcomings, including duplicate entries, were found in Germany,[26] indicating that they might characterise self-regulation more broadly.

Therefore, important gaps exist in our understanding of the accessibility and quality of European payment data. First, ongoing debates on the introduction of public regulation in some countries[5] suggest that the only comprehensive European regulatory overview[6] might have missed key regulatory developments, potentially with implications for data accessibility and quality.

Second, the implementation of the requirements of the EFPIA Code[19] has not been fully scrutinised. For example, although some trade groups will only meet the minimum standards (eg, by expecting companies to publish data on their websites), others might exceed them (eg, by creating centralised databases).[4 28] The need for establishing a comprehensive pattern of compliance is underscored by findings from Sweden and the UK, suggesting failure of self-regulation of drug marketing to meet some of its own key promises.[20 39]

Third, regulatory approaches in many European countries have escaped scrutiny,[4] making it unclear whether payment data reported in these countries share the strengths and weaknesses identified elsewhere. Consequently, although some aspects of government regulation, such as a greater scope of covered industries, have been demonstrated as superior to self-regulation,[4 6] it remains uncertain whether this is reflected by payment data accessibility or quality.[4]

Finally, the to-date evaluative criteria need refinement, as some, such as 'user friendliness', have attracted a contrasting appraisal of the same disclosure database by different expert commentators.[4 22]

We have two objectives. First, to identify regulatory approaches to payment disclosure in Europe. Second, to examine the accessibility and quality of payment data disclosed in countries with different approaches to disclosure.

## METHODS

### Data collection

#### Identification of regulatory approaches

To identify regulatory approaches to payment disclosure in Europe, PO and LM identified available peer-reviewed English-language research on the regulation of drug company payment disclosure. We searched Scopus using the terms 'Sunshine Act', 'Open Payments', as well as 'European Federation of Pharmaceutical Industries and Associations' and 'EFPIA', combined with 'disclosure'. We applied the same terms in the Google search engine to identify 'grey literature', including non-peer-reviewed reports.

Subsequently, PO and LM conducted iterative searches on websites dedicated to industry payment disclosure, including EFPIA's website and its national trade group members' websites. We also examined the country profiles published by MediSpend[40] and the websites of four major companies with presence across Europe (Amgen, GSK, Merck Serono, and Bayer), providing access to company disclosure methodologies, which reflect local regulatory requirements. Finally, we considered the websites of public or multistakeholder bodies which the previous steps identified as involved in overseeing payment disclosure.

Finally, PO surveyed industry trade groups and public or multistakeholder bodies overseeing payment disclosure (online supplemental file 1). The first round of standardised questions was emailed in mid-November 2020, followed up by reminder messages in late December 2020, asking recipients to provide answers by the end of the first week of January 2021. Of 34 approached pharmaceutical trade groups, 17 replied. Of those, 14 answered at least some of the questions, while the remaining ones sent holding messages. Of 13 approached public or multistakeholder bodies, 10 replied. Of those, six answered at least some of the questions, three sent holding messages and one redirected us to another institution (online supplemental file 2).

#### Data on accessibility and quality of payment disclosures

First, in countries with self-regulation, we considered industry codes, reports, press releases, trade group websites, and industry-run databases. Second, LM and P-AJ recorded their observations regarding the format and structure of payment data when designing scripts for scraping company and trade group websites to be integrated in eurosfordocs.eu.[28] Third, in countries with

disclosure overseen by public or multistakeholder bodies, the data included relevant legislation, the websites of bodies managing payment disclosure and disclosure databases. Fourth, in both countries with self-regulation and public regulation, we considered responses from our stakeholder survey. Finally, in countries with self-regulation and covered by eurosfordocs.eu, we collected—for verification purposes—national-level summary statistics published by EFPIA, industry trade groups and survey responses from the trade groups.

### Data analysis

#### Content analysis

Most of the source material was available in English. If this was not the case, we used Google Translate and Deepl. com, clarifying any linguistic issues by cross-checking with other online sources and consulting with relevant national bodies and colleagues with language expertise.

We coded the regulatory approaches deductively, building on an earlier categorisation which distinguished countries with self-regulation, government regulation and a combination of the two.[4] We modified it by considering new regulatory developments, such as the 2016 decision by the Spanish Data Protection Agency,[41] making disclosure by healthcare professionals compulsory without new government regulation.[28] Therefore, we replaced the 'government regulation' category with 'public regulation', comprising 'government regulation', that is, legislation relating directly to payment disclosure and 'regulatory intervention', that is, decisions by data protection agencies clarifying the rules of payment disclosure based on other existing legislation.

Deductive codes relating to data accessibility and quality were developed using earlier research.[4 5 24] Inductive coding was applied to the types of disclosed information and company techniques of decreasing data accessibility, which were identified when integrating industry data within eurosfordocs.eu.

The data were coded by PO and results were validated by team discussions, resolving any differences by agreement. In analysing industry-self regulation, we set the characteristics of disclosed data against recommendations from the EFPIA Code. Similar comparison was not necessary in relation to public regulation as it does not introduce any optionality.

#### Descriptive statistical analysis

As eurosfordocs.eu involved data extraction using disclosures published by individual companies and industry trade groups, we estimated the match between the database and the underlying data by comparing the value of payments calculated in specific countries using eurosfordocs.eu with national-level summaries obtained from industry sources.

#### Outcome measures

We had one primary outcome measure identifying the regulatory approaches to payment disclosure in each

country—self-regulation, public regulation and a combination of the two. As we identified both self-regulation and public regulation in some countries, we noted the number of regulatory approaches in each country—single (only self-regulation, public regulation or a combination of the two) or two (self-regulation and public regulation used in parallel).

In countries with self-regulation, we recorded whether it was based on the EFPIA Code, including shared payment, donor and recipient categories, or involved a distinct national industry code. For countries following the EFPIA Code, we specified whether trade groups were obliged to do so as EFPIA members or did this voluntarily as non-members.

Considering countries with public regulation, we distinguished those using government regulation, regulatory intervention or both. In countries with government regulation, we distinguished those introducing bespoke legislation focusing on payment disclosure or incorporating new provisions into existing pharmaceutical or medical device legislation. In countries where public and self-regulation were used in parallel, we recorded whether any overlap existed between the donors, recipients and payments covered by each approach.

In countries combining self- and public regulation, we denoted the form of both self-regulation and public regulation and how they were integrated.

The measures of accessibility and quality reflected the heterogeneity of payment data presentation. The basic measure of accessibility applied in all countries was whether it was disclosed on a centralised database or multiple websites. In addition, for countries with centralised databases, we created a 'heat map' aiding data synthesis and interpretation (table 1).

On top of the measures included in table 1, we had one additional measure of quality for eurosfordocs.eu as a database derived from payment disclosures published by drug companies and industry trade groups. We estimated the comprehensiveness of data extraction by comparing the value of payments available in eurosfordocs.eu with those reported separately in national-level industry data summaries. We set three arbitrary levels of match—exact (no difference between eurosfordocs.eu and summary industry data), close (difference between eursofrdocs.eu and industry data worth less than 10% of summary industry data) and low (difference exceeding 10% of summary industry data).

Finally, in countries with self-regulation but without centralised databases, we examined whether industry trade groups created gateways leading to disclosure documents, as recommended by EFPIA.[19] To illustrate challenges in data accessibility, we also generated lists of examples of, first, deviations from the EFPIA-recommended data presentation format ('EFPIA disclosure template')[19] and, second, the ways of presenting data, which decreased its accessibility.

## Patient and public involvement

We did not involve patient groups or the public. Our policy recommendations seek to increase public engagement with payment data by enhancing its accessibility and quality.

## RESULTS

We first map the regulatory approaches to payment disclosure in Europe. We then examine the accessibility and quality of payment data published by pharmaceutical companies and trade groups in countries with self-regulation. Subsequently, we focus on industry data in the subset of countries with self-regulation and covered by eurosfordocs.eu. Finally, we analyse payment data in countries with public regulation or combining public regulation with industry self-regulation.

### Mapping European regulatory approaches to payment disclosure

Before analysing the accessibility and quality of industry payment data, we must describe how it is disclosed in each European country (table 2).

We identified self-regulation in 30 countries in the form of codes of practice issued and overseen by industry trade groups.[19] In 28 of those, the industry codes incorporate the provisions of the EFPIA Code[19 42] as a necessary requirement of trade groups membership in EFPIA. This makes self-regulation, the 'default approach' to payment disclosure in Europe, with EFPIA holding power to exempt certain countries from following its Code.[43] The first exception is Luxembourg. While the Luxembourgish trade group is not an EFPIA member, it decides voluntarily to implement the regulation of payment disclosure modelled on the EFPIA Code.[44] The second exception is Denmark. Although the Danish trade group is an EFPIA member, EFPIA exempts Denmark from following its Code, given the country's separate public regulation provisions.[43] As the public regulation of payment disclosure in Denmark covers only healthcare professionals,[45] the Danish pharmaceutical trade group developed an additional code of practice covering only 'grants and donations' to hospitals.[46]

We found public regulation in 11 countries. In all cases, it takes the form of government regulation, in which provisions relating to payment disclosure are included either in bespoke new legislation (France, Lithuania and Romania) or are incorporated into existing pharmaceutical legislation (the remaining countries). In addition, in Greece, the Data Protection Agency made a regulatory intervention by issuing an interpretation of the government regulation.[47]

Only in France, Portugal and Turkey public regulation is the sole regulatory approach, replacing self-regulation entirely. EFPIA excepted France and Portugal from applying the EFPIA Code considering the nature of their public regulation[43]; however, the implementation of the

**Table 1** Heat map of measures of accessibility and quality of payment databases

**Measures of payment data accessibility**

| | | Higher accessibility | | Lower accessibility |
|---|---|---|---|---|
| Database format | How is the database published (ie, PDF, XLS, CSV, webpage)? | Webpage, XLS or CSV | Readable PDFs | Image-based PDFs |
| Database structure | Does the data from all companies follow a single template consistently? | Yes | N/A | No |
| Database searchability | Can the database be searched? If so, can database searches be carried out without data users providing any additional information? | Yes | Database searchable but additional information needed for searches | No |
| Customisable summary statistics | Does the database offer users the possibility of generating real-time, dynamic data summaries based on selected database characteristics? | Yes | N/A | No |
| Downloadability | Can the database be downloaded (eg, as a single CSV or XLS file) for further analysis? | Yes | N/A | No |

**Measures of payment data quality**

| | | Higher quality | | Lower quality |
|---|---|---|---|---|
| Spectrum of disclosed characteristics | What characteristics are included in relation to donors, recipients and payments? | All characteristics from the EFPIA disclosure template covered as well as some additional ones | All characteristics from the EFPIA disclosure template covered | At least some characteristics from the EFPIA disclosure template not covered, including instances where some additional characteristics are provided |
| Aggregation of payments | Are payments itemised (ie, all payments have separate entries) or are they aggregated on an annual basis (eg, per recipient and/or payment category)? | All payments itemised | Some payments itemised, others aggregated | All payments aggregated |
| Inclusion of taxes | Is it clear whether payments are reported inclusive or exclusive of any taxes, such as VAT? | Single rule for all companies and payments | No single rule, each company sets its own rules for VAT reporting which are published separately from payment disclosures* | Rules around tax reporting are unclear |
| Unique identifiers | Do reported donors (drug companies) or recipients (healthcare professionals or organisations) have unique identifiers? | All donors and recipients | Some donors or recipients | No unique identifiers |

*The EFPIA Code stipulates that companies must publish documents, called 'methodological notes', which should explain their approach to reporting VAT and other taxes. Companies publish these documents separately from payment disclosures but consulting them is necessary to understand, compare and aggregate payment values.
CSV, comma-separated values files ; EFPIA, European Federation of Pharmaceutical Industries and Associations; PDF, portable document format; VAT, Value-Added Tax; XLS, spreadsheet file format used in Microsoft Excel.

EFPIA Code in Turkey is only suspended while its compatibility with the EFPIA Code is being reviewed.[48]

In the remaining eight countries with public regulation, there is also parallel self-regulation. In four of these (Denmark, Lithuania, Romania and Slovakia), self and public regulations cover different donors, payments or recipients, whereas in the remaining ones (Estonia, Greece, Hungary and Latvia), donors, recipients and payments disclosed via public and self-regulation may overlap. Consequently, the existence of parallel self and public regulation in the eight countries means that

self-regulation is used exclusively in 22 of the 30 countries with this approach.

Self- and public regulation are combined as a single approach in four countries. Contrasting with countries with public regulation, here, the industry contributes to managing payment disclosure. However, unlike in countries with self-regulation, the industry derives at least some of its regulatory power from public authorities, often sharing it with other stakeholders. In two of the four countries, public regulation takes the form of government regulation (Belgium and Finland), and, in

**Table 2** Approaches to regulating payment disclosure in European countries

| Country* | Regulatory approaches to payment disclosure | | |
| --- | --- | --- | --- |
| | Self-regulation | Public regulation | Combination of self-regulation and public regulation |
| Austria | ✓ | | |
| Bosnia and Herzegovina | ✓ | | |
| Bulgaria | ✓ | | |
| Croatia | ✓ | | |
| Cyprus | ✓ | | |
| Czech Republic | ✓ | | |
| Germany | ✓ | | |
| Iceland | ✓ | | |
| Ireland | ✓ | | |
| Italy | ✓ | | |
| Luxembourg | ✓ | | |
| North Macedonia | ✓ | | |
| Malta | ✓ | | |
| Norway | ✓ | | |
| Poland | ✓ | | |
| Russia | ✓ | | |
| Serbia | ✓ | | |
| Slovenia | ✓ | | |
| Sweden | ✓ | | |
| Switzerland | ✓ | | |
| UK | ✓ | | |
| Ukraine | ✓ | | |
| Denmark | ✓ | ✓ | |
| Estonia | ✓ | ✓ | |
| Greece | ✓ | ✓ | |
| Hungary | ✓ | ✓ | |
| Latvia | ✓ | ✓ | |
| Lithuania | ✓ | ✓ | |
| Romania | ✓ | ✓ | |
| Slovakia | ✓ | ✓ | |
| France | | ✓ | |
| Portugal | | ✓ | |
| Turkey | | ✓ | |
| Belgium | | | ✓ |
| Finland | | | ✓ |
| The Netherlands | | | ✓ |
| Spain | | | ✓ |
| n=37 | n=30 | n=11 | n=4 |

*Excluded countries: Albania, Andorra, Belarus, Lichtenstein, Monaco, Montenegro, San Marino and Vatican City.

the two others—regulatory intervention (Spain and the Netherlands).

Belgium regulates payment disclosure via a bespoke 'Sunshine Act', but the interpretation of its key provisions is left to betransparent.be, a multistakeholder body involving industry and professional organisations,[49 50] which also runs the Transparency Register integrating company disclosures.[51 52] In Finland, new provisions have been introduced into the Medicines Act, stipulating that drug companies 'must keep available for public review' a list of all payments to 'associations in the fields of medicine and healthcare',[53] but, in practice, the disclosure takes place following the EFPIA Code.

In Spain, public regulation involves an intervention by the Data Protection Authority,[41] confirming that the publication of named payment recipients does not require recipient consent.[28] However, like in Belgium and Finland, disclosure is managed by companies based on the EFPIA Code. In the Netherlands, payments are disclosed using self-regulatory rules developed by the Foundation for the Code for Pharmaceutical Advertising, which are separate from the EFPIA Code. Like in Belgium, the central platform is a multistakeholder body involving, in this case, the industry and healthcare providers.[54] However, public authorities triggered the policy debate on payment disclosure and, having considered self-regulation preferable to public regulation, they lent it financial support and monitor its performance.[55 56] Furthermore, consistent with the regulatory intervention in Spain, the Dutch Data Protection Authority confirmed that recipient consent is not required for payment disclosure.[56]

## Data disclosed via self-regulation by pharmaceutical companies and trade groups

We were able to collect information on accessibility and quality of payment data in 28 of the 30 countries with self-regulation.

Regarding data accessibility, the EFPIA Code allows companies within each country to disclose payments either on a centralised platform or individual websites.[19] However, only five trade groups have established databases for all companies, including four countries following the EFPIA Code and one using its own code (Danish Association of the Pharmaceutical Industry, LIF). Of the five industry-run databases, none had customisable summary statistics (table 3). Moreover, only one was fully searchable (ie, without additional information required for searches) and just two were downloadable. Overall, Disclosure UK had by far the highest data accessibility.

Turning to data quality, only the Czech database used unique donor and recipient identifiers consistently, but, because they were required for searches, they paradoxically decreased data accessibility. The second most frequent problem across the databases was tax reporting. While in the four databases established under the EFPIA Code, the rules on tax reporting might be reconstructed using 'methodological notes' published separately by each company,[19] the Danish database had no information

**Table 3** Accessibility and quality of drug company payment data disclosed via centralised industry databases and eurosfordocs.eu

| Country | Name of regulation | Overseeing authority and database web link[2-4] | Payment data accessibility* | | | | | | Payment data quality* | | | |
| | | | Document format | Single data template | Database searchable | Customisable summary statistics | Database downloadable | Characteristics included | Aggregation of payments | Payments with or without taxes | Unique identifiers |
| **Self-regulation at the European level—minimum requirements** | | | | | | | | | | | |
| EFPIA | EFPIA Code | EFPIA | Not regulated | Yes ('EFPIA disclosure template'), but deviations allowed | Not regulated | Not regulated | Not regulated | Donors; recipients; recipient location; payment categories and amounts; year | Annually per payment type | No single rule, each company sets its own rules for VAT reporting which are published separately from payment disclosures | Optional |
| **Centralised online industry databases** | | | | | | | | | | | |
| UK | ABPI Code of Practice | Association of the British Pharmaceutical Industry | Website, XLS | Yes | Yes | No | Yes | Donors; recipients; recipient categories (healthcare professionals) and location; payment categories and amounts; year; web links with further descriptions for some payments | Annually per payment type for healthcare professionals; payments to healthcare organisations itemised | No single rule, each company sets its own rules for VAT reporting which are published separately from payment disclosures | No |
| Czech Republic | Eticky Kodex AIPF | Asociace inovativního farmaceutického prûmyslu | Website | Yes | Yes (but requires donor or recipient identifiers) | No | No | Donors; donor location; recipients; recipient location; payment categories and amounts; year | Annually per payment type | No single rule, each company sets its own rules for VAT reporting which are published separately from payment disclosures | Recipient and donor identifiers |
| Denmark | Ethical rules for the pharmaceutical industry's donations and grants | Laegemiddelindustriforeningen | Readable PDFs | Yes | No | No | Yes | Donors; project name; recipients; product name; funded activity; payment goal; timescale of funded activity; payment amount and form (cash or benefit in kind) | No | Unclear | No |
| Greece | SFEE Code of Conduct | Hellenic Association of Pharmaceutical Companies | Website | Yes | No | No | No | Donors; recipients; recipient categories; payment descriptions, categories goals, and amounts; date | No | No single rule, each company sets its own rules for VAT reporting which are published separately from payment disclosures | No |
| Ireland | Code of Practice of the Pharmaceutical Industry | Irish Pharmaceutical Healthcare Association | Website | Yes | No | No | No | Donors; recipients; recipient location; payment categories and amounts; year | Annually per payment type | No single rule, each company sets its own rules for VAT reporting which are published separately from payment disclosures | Partial (recipient identifiers used by some companies) |

Continued

**Table 3** Continued

| Country | Name of regulation | Overseeing authority and database web link[§] | Payment data accessibility* | | | | | Payment data quality* | | | |
|---|---|---|---|---|---|---|---|---|---|---|---|
| | | | Document format | Single data template | Database searchable | Customisable summary statistics | Database downloadable | Characteristics included | Aggregation of payments | Payments with or without taxes | Unique identifiers |
| **Industry data integrated within an independent database†** | | | | | | | | | | | |
| Euros fordocs.eu | Codes of conduct in countries where data was collected‡ | N/A | Website, XLS | Yes | Yes | Yes | Yes | Donors; recipients; recipient location; payment categories and amounts; year | Annually per payment type for healthcare professionals (all countries). Annually per payment type for healthcare organisations in all countries but the UK, where payments to healthcare organisations are itemised | No single rule, each company sets its own rules for VAT reporting which are published separately from payment disclosures | Spain: recipient identifiers. Other countries: No |

*Lighter colours indicate, respectively, higher, and darker colours—lower, data accessibility and quality. In the upper part of the table, the centralised industry databases are presented in the descending order of their overall data accessibility and quality, that is, the greater overall number of lighter cells a database has the higher its position within the table. Databases with equal numbers of lighter and darker cells are sorted alphabetically.
†The disclosure requirements ordinarily cover both healthcare professionals and organisations. The exceptions are the database run by the Danish pharmaceutical industry trade group (donations to hospitals) and the database run by the Greek pharmaceutical industry trade group (only payments to healthcare organisations).
‡Ireland, Italy, Germany, Spain, Sweden, Switzerland and the UK.
§Web links are accurate as of May 2021.
¶Some pharmaceutical industry trade groups create and delegate some responsibility for the everyday operation of their codes to sub-divisions such as the Ethical Committee for the Pharmaceutical Industry (established by Denmark's LIF) or the Prescription Medicines Code of Practice Authority (established by the UK's ABPI). However, the ultimate responsibility for managing and overseeing the codes is with the trade group.
EFPIA, European Federation of Pharmaceutical Industries and Associations; VAT, Value-added tax.

regarding tax. Taken altogether, Disclosure UK had the highest data quality, although here it was more closely matched by the Czech database.

In 23 of the remaining countries with self-regulation and available data, disclosures were published on individual websites for each company. Of these, in 18 countries, trade groups had the EFPIA-recommended gateways to these websites.[19 23] Nevertheless, without EFPIA's explicit guidance on the electronic format of disclosure documents, disclosures published on company websites in countries with and without gateways were typically PDFs. While some of these documents were 'readable', allowing for copying and pasting of information, they offered limited possibilities for efficient searches and integrating data from different companies. Additionally, some companies presented data without strictly following the 'EFPIA disclosure template',[19] which further impeded possibilities for cross-company comparisons (online supplemental file 3 has examples of these deviations). Some firms apparently manipulated data presentation using low-resolution, image-based PDFs, which prevented any searches (online supplemental file 4 summarises these techniques).

Given the low accessibility of payment data, analysing its quality was practically impossible in countries without centralised databases. Therefore, we do this using eurosfordocs.eu, a database covering drug company disclosures in countries with self-regulation (Ireland, Italy, Germany, Sweden, Switzerland and the UK); in this part of the analysis, we also include Spain, a country with a combination of self- and public regulation, as it helps illustrate problems characteristic of self-regulation.

### Industry data disclosed via self-regulation and integrated within eurosfordocs.eu

As already demonstrated in the previous section, Eurosfordocs.eu had data accessibility superior to all industry-run databases (table 3). While the Irish and UK databases were also searchable, eurosfordocs.eu offered customisable queries using combinations of donor and recipient names and payment categories.[57] It was the only database offering customisable summary statistics enhancing data exploration. In addition, only eursofordos.eu and Disclosure UK were downloadable for further analysis.

A specific consideration regarding data integrated within eurosfordocs.eu is estimating how closely they match the underlying industry disclosures (table 4). Complete data extraction was only possible in the UK and Ireland, the two countries with centralised trade group databases (online supplemental file 5 summarises the data extraction statistics). Elsewhere, data scraping prioritised the 20 largest donors known from the countries with complete data; more data were scraped whenever allowed by formats used by companies.[28] For four of the six countries, the resulting data set closely or exactly matched the industry's summary country-level data. The two countries with a low match were Germany and Spain,

**Table 4** Estimation of the comprehensiveness of industry payment data extracted for eurosfordocs.eu (2019)

| Country* | Total value of payments reported in summary industry data (€m)[2 3] | Total value of payments extracted to eurosfordocs.eu (€m)[4 5] | Difference (€m) | Difference as a share of summary industry data (%)† | Level of match between summary industry data and eurosfordocs.eu |
| --- | --- | --- | --- | --- | --- |
| Germany | 629 | 499 | 130 | 21% | Low |
| Ireland | 35 | 35 | 0 | 0% | Exact |
| Sweden | 90 | 82 | 8 | 9% | Close |
| Switzerland | 167 | 155 | 12 | 7% | Close |
| Spain | 601 | 337 | 264 | 44% | Low |
| UK | 619 | 611 | 8 | 1%‡ | Exact/close |

*Only countries covered by both eurosfordocs.eu and available national-level summary data generated by industry trade groups are included.

†Some of the difference between the value of payments based on summary industry data and extracted to eurosfordocs.eu results from the differences in the exchange rates. This is exemplified by the examples of Ireland (both values in euro, no difference) and the UK (original values in the sterling, the difference is caused by different exchange rates used to convert the sterling to euro). By contrast, the 1% difference between eurosfordoscs.eu and Disclosure UK results from two marginally different exchange rates used to convert the sterling to euros.

‡All payment values in non-euro currencies were converted to euros based on the exchange rate obtained from the CurrencyConverter,[85] a Python library for exchange rates.

§Sources of national-level summary payment data. (a) Germany,[86] Spain,[87] Switzerland[88]—publicly available pharmaceutical industry summary data published by the pharmaceutical industry trade groups. (b) Ireland—a combination of an Europe-wide report published by EFPIA[89] and email communication with the Irish pharmaceutical industry trade group.[90] (c) Sweden—email communication with the pharmaceutical industry trade group. (d)The UK—calculations based on data obtained from Disclosure UK, the centralised database of industry payments run by the Association of the British Pharmaceutical Industry.[91]

¶All payment values in non-euro currencies were converted to euros based on the average yearly exchanged rates published by the European Central Bank.

**The source of payment values reported in this column are centralised pharmaceutical industry payment databases (Ireland and the UK) and payment reports covering payments made by individual companies (Germany, Spain, Sweden and Switzerland).

given a high proportion of image-based PDFs hindering data extraction.[28]

Nevertheless, other aspects of the data quality in eurosfordocs.eu share key limitations with the underlying company disclosures.

First is a narrow spectrum of reported recipient, donor and payment characteristics. Eurosfordocs.eu does not present payment distribution within the healthcare system due to the incoherent use or omission of recipient categories by drug companies. Of all countries covered by eurosfordocs.eu, the UK is the only one where the industry trade group categorised healthcare professionals receiving payments,[58] although incoherently[21]; healthcare organisations were nowhere categorised.

Second, consistent with the EFPIA Code[19] payments to healthcare professionals are not itemised but aggregated annually per recipient within each payment category. The same applies to payments to healthcare organisations, except for the UK, where the ABPI mandates that payments to healthcare organisations be itemised.[58] This UK-specific rule might explain the large difference in the number of payments reported with Germany, a country with a similar overall value of payments (online supplemental file 5). However, it is equally possible that not all companies in the remaining six countries covered by the database aggregate payments consistently as some list more than one payment per recipient, which might also indicate that although these recipients have the same names, they are different entities.

Third, the reported payment values must be interpreted cautiously as it is unclear whether they include taxes without consulting the separately published 'methodological notes'.[19] Some companies have different approaches to tax reporting depending on payment or recipient categories. Consequently, establishing the value of payments made by each company requires additional forensic work.[24]

Finally, while EFPIA introduces the option of unique recipient identifiers in disclosed payment data,[19] of the seven countries covered by eurosfordocs.eu only the Spanish trade group followed this recommendation. Elsewhere the number of recipients per company and, consequently, the value of payments per recipient remains unknown. Given inconsistent naming approaches in disclosures made by the same or different companies, the same recipient can have different names, and, conversely, different recipients may have the same name.[24] Furthermore, the same recipient can be identified at different levels of aggregation (eg, hospital wards, departments or hospitals), with self-regulation at least in some countries placing the onus of identifying possible multiple records on payment recipients and not companies.[24 59] Finally, without identifiers, payment data cannot be connected to other databases.

### Data disclosed via public regulation or a combination of public and self-regulation

Having examined countries with self-regulation, we proceed to those with public regulation or a combination of public and self-regulation.

Of the 15 countries with public regulation or a combination of self-and public regulation, all but two had centralised databases. The exceptions were Finland and Spain, where disclosures were made on individual drug company websites, consistent with the EFPIA Code. Of the 13 countries with centralised databases, one had a database, which was not publicly available (Turkey) and two others had separate databases for different payment categories (Denmark) and healthcare professionals and organisations (Greece). As the information included in the separate Danish and Greek databases, it did not differ according to our outcome measures, we consider them jointly (table 5).

The databases established via public regulation or a combination of public and self-regulation had the pattern of accessibility similar to the industry-run databases. Of the 13 databases, none had customisable summary statistics, and only 6 were downloadable and fully searchable. Overall, Transparence Santé was the frontrunner.

The most frequent data quality shortcoming was unclear tax reporting, with only two databases providing relevant rules. However, over half of the databases had at least partial donor or recipient identifiers, which was the most frequent problem in the industry-run databases. Furthermore, just five databases covered a spectrum of donor or recipient characteristics exceeding the minimum recommendations from the EFPIA Code . Transparence Santé again had the highest overall data quality.

In sum, Transparence Santé had combined data accessibility and quality exceeding that of Disclosure UK, the frontrunner industry database.

## DISCUSSION

Our policy review suggests that payment data disclosure does not automatically increase transparency of financial relationships between drug companies and the healthcare sector.[4 5] Consistently with research on disclosure of aspects of health policymaking by both public and private-sector actors, we find that achieving 'practical' or 'actionable' transparency is no less important than introducing transparency rules themselves.[60–62]

Although EFPIA calls payment data generated via self-regulation 'open to public scrutiny',[63] establishing the entanglement of any recipient, let alone a system-level picture, is impossible given the dispersal of disclosures on company websites in most European countries. Additionally, documents published as PDFs, sometimes in ways suggesting deliberate attempts to impede user engagement, fall below the Australian industry-endorsed regulations, requiring firms to use an analysable format.[5] Therefore, self-regulation cannot address 'the issues of perceived conflict of interest',[64] as promised by EFPIA. More broadly, the evidence of some companies and trade groups meeting only the minimum requirements from the EFPIA Code, or fulfilling them in ways inconsistent with the Code's spirit, reflects the limited success of self-regulation in modifying corporate behaviour in areas of

public health policy such as reduction of sugar content in food[65] or managing viewers' exposure to alcohol advertising.[66]

EFPIA is clearly aware of at least some of the problems in payment data accessibility. For example, in 2019, it listed 'improv(ing) access' via '[c]reateing platforms with [a] searchable tool' as one of the 'main topics' to be considered by EFPIA itself and its member trade groups.[23] However, little evidence exists of subsequent discussions on this issue except for a planned 'feasibility study' of possible 'options for improving the disclosure' to be considered from 2021 to 2023.[67] Furthermore, EFPIA does not seem to have recognised or engaged with the issues of low payment data quality.

Against this background, eurosfordocs.eu radically enhances data accessibility in countries without centralised industry databases, also enabling comparative investigations of country payment patterns,[28] which is important given the accelerating EU-wide health initiatives.[68] Although the customisable opportunities for data exploration are new to the public, data analytic firms have offered them as a consultancy service to drug companies.[40 69] Consequently, eurosfordocs.eu may contribute to changing what may be the de-facto status of payment data as a commodity used to monitor internal compliance with disclosure requirements and potentially inform marketing strategies targeting healthcare professionals.[70]

In countries with self-regulation, the challenges in data accessibility and quality are exacerbated by nondisclosed payments. EFPIA admits the problem of '[c]onsent issues in general but also by country and by specialty',[23] while evidence also exists of varying consent rates between companies.[28] In addition, some companies may not disclose all their payments, as suggested by instances of underreporting of payments to patient organisations, with their disclosure also regulated by the EFPIA Code but with distinct policies.[71 72] Furthermore, self-regulation only covers companies and trade groups that have ratified the EFPIA Code or its transposition into country-level codes. Therefore, disclosure requirement may not extend to companies focusing on generic or over-the-counter medicines and even major manufacturers of branded prescription medicines (eg, Vertex does not follow the ABPI Code). However, some non-member companies may choose to follow the trade group codes voluntarily. For example, the list of Disclosure UK participants exceeds ABPI membership.[72] Furthermore, some companies may belong to other trade groups (eg, generic or small biotech trade groups), which, in some countries, require their members to abide by the national codes (eg, Sweden, Denmark). Problems with underreported payments may be particularly prominent in countries with parallel self and public regulation due to possible confusion relating to where payments should be reported. For example, some healthcare organisations in England underreported some of the payments they had received given their implicit or explicit expectations that the payments would be disclosed via self-regulation.[30 31]

**Table 5** Accessibility and quality of drug company payment data disclosed via public regulation or a combination of self-regulation and public regulation

| Country | Name of regulation | Overseeing authority and database web link[2-4] | Payment data accessibility* | | | | | Payment data quality* | | | |
|---|---|---|---|---|---|---|---|---|---|---|---|
| | | | Document format | Single data template | Database searchable | Customisable summary statistics | Database downloadable | Characteristics included | Aggregation of payments | Payments with or without taxes | Unique identifiers |
| France | Law Number 2011–2012 (Law Bertrand) | Ministry of Social Affairs and Health | Webpage | Yes | Yes | No | Yes | Donors; donor categories; recipients; recipient categories; payment categories and amounts; date; recipient address | No | Inclusive of VAT | Donors (multiple entries for subsidiaries), recipients (partial) |
| Latvia | Regulation Number 378 (2014) | Health Inspectorate | XLS | Yes | No | No | Yes | Donors; recipients; recipient categories; payment name, description, category and amount; date; recipient address | No | Unclear | Donors, recipients |
| Belgium | Sunshine Act of 2016 | Federal Agency for Medicines and Health Products | Webpage | Yes | Yes | No | No | Donors; recipients; recipient categories; payment categories and amounts; recipient address; years | Annually per payment type | Unclear | Donors, recipients |
| Lithuania | Law on Pharmacy (provisions from 2019), Ministerial Order Number V-1537 (2020) | State Medicines Control Agency | XLS | Yes | No | No | Yes | Donors; recipients; recipient categories; payment name; date; recipient address | No | Unclear | Donors (not publicly available), recipients (publicly available) |
| Portugal | Decree Law 20/2013 and 128/2013 | National Authority of Medicines and Health Products | Webpage | Yes | Yes | No | No | Donors; donor categories; recipients; payment descriptions and amounts; years | No | Inclusive of VAT | No |
| Romania | Orders of the Minister of Health 194/2015 and 874/2015 | National Agency for Medicines and Medical Devices | Webpage | Yes | Yes | No | No | Donors; recipients; recipient categories; payment descriptions, categories, and amounts; recipient address; date | No | Unclear | No |
| Slovakia | Act Number 362/2011 on Medicines and Medical Devices | National Health Information Center | XLS | Yes | No | No | Yes | Donors; recipients; recipient categories (only healthcare professionals); payment descriptions, categories and amounts; clinical trial numbers; product names; recipient address; date | No | Unclear | No |

Continued

**Table 5** Continued

| Country | Name of regulation | Overseeing authority and database web link[2-4] | Payment data accessibility* | | | | | Payment data quality* | | | |
|---|---|---|---|---|---|---|---|---|---|---|---|
| | | | Document format | Single data template | Database searchable | Customisable summary statistics | Database downloadable | Characteristics included | Aggregation of payments | Payments with or without taxes | Unique identifiers |
| Denmark | Health Act of 2014, Executive Order Number 1153 | Danish Medicines Agency 1. Conferences abroad; 2. Professional affiliations | Webpage | Yes | Yes | No | No | Conferences abroad—donors; recipients; recipient categories; recipient address; Professional affiliations—donors; recipients; recipient categories; recipient address; payment amounts | Annually per payment type | Unclear | Recipients |
| Hungary | Act XCVIII of 2006 (provisions introduced in 2011) | National Institute of Pharmacy and Nutrition | Webpage | Yes | Yes | No | No | Donors; payment names, descriptions and amounts; date; recipient address | No | Unclear | No |
| The Netherlands | Code of Conduct for Pharmaceutical Advertising (2012) | Foundation for the Code for Pharmaceutical Advertising | Webpage | Yes | Yes (recipient identifiers needed) | No | No | Donors; recipients; recipient categories; payment categories and amounts; year | Annually per payment type | Unclear | Recipients |
| Greece | Law 4316/2014; Opinion Number 5/2016 and 2/2017 of the Data Protection Authority; circular number 17770/2016 of the National Authority for Medicines | National Organisation for Medicines 1. Payments to conference participants 2. Payments to conference organisers; drug company websites | PDFs – image-based | Yes | No | No | Yes | Payments to conference participants—donors; recipients; payment categories (types of conference expenditure) and amounts; year; Payments to conference organisers—donors; payment amounts; year | Payments to conference participants—annually per recipient; Payments to conference organisers—donors | Unclear | Donors |
| Estonia | Medicinal Products Act of 2005 (provisions introduced in 2013) | State Agency of Medicines | XLS | Yes | No | No | Yes | Donors; payment categories and amounts; payment location (country); year | Annually per donor | Unclear | No |
| Turkey | Regulation on Promotional Activities of Medicinal Products for Human Use 2015 | Ministry of Health (database not publicly available) | Unclear | Unclear | Unclear | Unclear | Unclear | Unclear | Unclear | Unclear | Unclear |

*Lighter colours indicate, respectively, higher, and darker colours—lower, data accessibility and quality. The databases are presented in the descending order of their overall data accessibility and quality, that is, the greater overall number of lighter cells a database has the higher its position within the table. Databases with equal numbers of lighter and darker cells are sorted alphabetically.
†This column provides the dates when public regulation of payment disclosure was first introduced. If public regulation of payment disclosure forms part of a larger piece of government regulation, it is specified—where appropriate—whether the regulation of payment disclosure was introduced as a change already existing government regulation. The dates reported here do not cover changes to or refinements of provisions focusing on payment disclosure.
‡The disclosure requirements ordinarily cover both healthcare professionals and organisations. The exceptions are the Danish databases (only healthcare professionals) and the Turkish database (it is unclear whether disclosure requirements also cover healthcare organisations).
§Web links are accurate as of May 2021.
¶The recipient addresses ordinarily refer to the location of the payment recipient. In the case of Hungarian, Latvian and Lithuanian databases we considered that the event addresses were equivalent to recipient addresses.

Although data reported in the US Open Payments database have attracted some criticism,[25 73] data accessibility and quality are vastly superior to European data disclosed using not only self-regulation but also public regulation. While the example of Transparence Santé indicates that public regulation can generate payment data outpacing industry-run databases, it often shares major shortcomings with self-regulation, including the lack of recipient identifiers or payment itemisation.[24 26] Moreover, in some databases, the spectrum of disclosed characteristics is even narrower than the minimum which EFPIA recommends for the industry. Nevertheless, public regulation eliminates optionality characterising the EFPIA Code regarding, for example, centralised databases. The legally binding nature of public regulation should also involve high levels of compliance. However, instances of inaccurate or incomplete reporting by some companies are possible.[25]

Inconsistencies in the approaches to public regulation between European counties are highlighted by EFPIA and used as a key argument in favour of self-regulation, which, in EFPIA's words, represents a 'global and consistent approach for companies across Europe and common understanding for the public'.[23] France is one country in which problems in data accessibility and 'ergonomics' have been recognised by the Ministry of Health in 2018.[74] Following this, a new version of Transparence Santé is due to be launched in late 2021 and is expected to adopt approaches to data presentation, including visualisations, similar to those developed earlier for eurosfordocs.fr. We are not aware of similar discussions in other countries with public regulation or combining self- and public regulation.

Therefore, one key area of further study would involve using qualitative methods to identify and trace relationships between the likely causes of limited corrective action seeking to address the shortcomings of the current reporting systems in European countries. Of particular importance would be examining the incentive structures and motivations of public authorities, industry trade groups and companies, healthcare professional associations as well as patient organisations at the national and EU levels.

## Limitations

This study has several limitations. Our measures of data accessibility could be expanded. For example, some databases are difficult to find, including web links to the Greek and Latvian databases published within news releases, without permanent online location. Similarly, although Transparence Santé can be downloaded, the size of the data set prevents it from being opened using the standard Microsoft Excel package. Data quality could be scrutinised further by considering the types of disclosed donors, payments and recipients.[4] Furthermore, qualitative insights from data users would be essential for ranking the outcome measures and attributing weights to their values, such as degrees of user friendliness.

Our focus on the database level might obscure cross-company differences. For example, the widely ranging consent rates achieved by companies from healthcare professionals suggest that similar differences can occur in data quality and accessibility.[21 28] Furthermore, we did not calculate company-level aspects of data accessibility (eg, the share of image-based PDFs) and quality (eg, the share of duplicate entries, consistency in using donor or recipient categories and identifiers, missing data and mistakes, such as negative values). Undertaking these calculations would have necessitated extensive forensic work.[24] However, these problems are likely to be widespread and serious, affecting even Transparence Santé, the database we ranked the highest based on its quality .[21 24 26]

## Conclusions and policy recommendations

We formulate suggestions for enhancing public engagement with disclosed payment data (table 6), which are also relevant for non-European countries, such as Japan, experiencing problems similar to those identified in this study.[75]

Payment data accessibility can be enhanced with only minor revisions of the existing regulatory approaches, with the top priority being centralised databases offering possibilities for payment exploration and contextualisation.

Improving payment data quality would require new comprehensive public regulation, preferably at the European level.[4 28] Following the example of the US Open Payments database, payments should be reported together with information on related products to allow exploring company marketing strategies.[24 76] Another vital piece of information to include might be the numbers of clinical trials associated with payments, as exemplified by the database run by the Slovak National Health Information Center. Furthermore, granular disclosure is vital for capturing payments of different sizes, with some US studies suggesting that even small payments impact prescribing behaviour,[77 78] while others indicating a more complex dose–effect relationship.[11 29 79 80] Data interpretation can be enhanced by descriptions of funded activities (eg, specific conferences or projects), consistent with the EFPIA Code's requirements regarding payments to patient organisations.[72 81] Recipient characteristics should be also expanded, reflecting how the public engages with the healthcare system.[24] Finally, Open Payments highlights that recipient identifiers are necessary for reliable analysis and connecting payment data to data sets with details of prescription and procurement.[11 29 77 78 80] Data integration and management require strong compliance mechanisms, including penalties for providing data of inadequate quality.[25]

Additionally, in European countries with self-regulation, eliminating possibilities for refusing disclosure by recipients is necessary to reduce high levels of missing data.[28] The decision by the Spanish Data Protection Authority is illustrative here, exempting payment data from the provisions of the European data protection legislation (the General Data Protection Regulation, GDPR).[28]

| | Table 6 How can public authorities and the pharmaceutical industry improve the transparency of payment data? |
|---|---|
| | **Recommendations for improving accessibility of payment data** |
| 1 | Create national-level databases searchable for companies, recipients and payment categories. |
| 2 | Make the databases in the CSV or XLS format for further analysis, while ensuring that the released data can be split using different variables, for example, by year or recipient type to make it manageable for users. |
| 3 | Enable users to explore the data by allowing them to generate data summaries placing payments made or received in a broader context (eg, payments made by other companies or received by the same or other recipient categories, such as medical specialty). |
| | Recommendations for improving quality of payment data. |
| 4 | Publish unique identifiers for payment recipients shared by all companies and used consistently over time. |
| 5 | Introduce clear rules on the levels of aggregation for identifying recipients (eg, clinic, ward or hospital) to enhance the consistency of reporting. |
| 6 | Introduce categories of recipients to enable mapping the distribution of payments in the healthcare system. The categories relating to healthcare professionals could include a standardised list of medical specialties. The categories covering healthcare organisations could reflect their functions in the healthcare system as providers, commissioners or professional organisations. |
| 7 | State clearly whether reported payments should include VAT or other taxes so that payment values from different companies can be compared reliably. |
| 8 | Publish each payment individually instead of aggregating them annually per recipient. |
| 9 | Publish payment descriptions so that the public can understand the activities they fund as well as their context. This requirement would follow the self-regulatory rules existing in relation to the disclosure of payments to patient organisations. |
| 10 | Enforce and publish detail of data quality checks: eliminate missing values, payments with the value of zero and ensure that each recipient has a unique name and is reported at the same level of aggregation by all companies. Other data quality checks should involve cross-checking recipient name and address information to ensure consistency and avoid duplicate reporting. |

CSV, comma-separated values files ; VAT, Value-Added tax; XLS, spreadsheet file format used in Microsoft Excel.

Finally, transparency alone cannot address FCOIs. Even the increased transparency brought in by Open Payments does not seem to have decreased physicians' acceptance of FCOIs or increased patient concerns about their possible effects on the care they receive.[82] Paradoxically, transparency may normalise FCOIs or increase their impact via moral licensing.[82] Therefore, transparency should be accompanied by policy measures seeking to reduce or eliminate certain FCOIs. Key European examples include banning some financial relationships,[83] including payments to healthcare professionals for conference participation in Sweden[28] or prohibiting sponsored meals over €60 in France.[84]

**Acknowledgements** We would like to express gratitude to all pharmaceutical industry trade groups and public authorities, which have responded to our request for information and were willing to explain the nature of local regulatory solutions. We thank, in particular, the ABPI's Disclosure UK Team for their continued interest in our research as well as professionalism and openness in sharing the fine detail of their important work. We extend our thanks to colleagues who took the time to read drafts of this paper or offer linguistic assistance: Marcell Csanadi, Eniko Csanadi, Alice Fabbri, Kevin Jean, Fatma Korkmaz, Olga Loblova, Savvas Morris, Emily Rickard, Eszter Saghy, Christos Vasilakis, and Weronika Ozieranska. We are also thankful to the BMJ Open Editors and three reviewers for their helpful and constructive comments.

**Contributors** PO is a Senior Lecturer at the Department of Social and Policy Sciences, University of Bath. PO conceived and wrote the paper, collected and analysed the data. PO is the guarantor of the paper. LM is a data scientist and the President of the Euros for Docs Association. LM created the eurosfordocs.eu database, analysed the data and contributed to writing. P-AJ formerly presided the Euros for Docs Association. P-AJ collaborated with LM on creating eurosfordocs.eu. P-AJ conceived the paper and contributed to writing. SM is an Associate Professor at the Department of Sociology, Lund University. SM conceptualised the paper and contributed to writing.

**Funding** This work was supported by The Swedish Research Council (VR), grant number 2020-01822 ('Following the money: cross-national study of pharmaceutical industry payments to medical associations and patient organisations').

**Competing interests** We have read and understood the BMJ Group policy on declaration of interests and declare the following interests: PO's PhD student was supported by a grant from Sigma Pharmaceuticals, a UK pharmacy wholesaler and distributor (not a pharmaceutical company). The PhD work funded by Sigma Pharmaceuticals is unrelated to the subject of this paper. LM and PAJ are members of Euros for Docs, a non-profit organization registered in France that seeks to promote transparency of drug company funding in the healthcare sector by making payment data accessible and complete across Europe. PAJ is employed by Haute Autorité de Santé, the French independent health technology assessment organisation. SM's partner is employed by PRA Health Sciences, a global Contract Research Organization whose customers include many pharmaceutical companies.

**Patient consent for publication** Not applicable.

**Ethics approval** No ethical approval was needed. The ethical implications of this study article were approved via a peer ethics review process at the Department of Social and Policy Sciences, University of Bath in February 2020. This study did not require a full ethical approval as it relied on publicly available data aggregated at the organisational or country level.

**Provenance and peer review** Not commissioned; externally peer reviewed.

**Data availability statement** All data relevant to the study are included in the article or uploaded as supplementary information. We have included all relevant data as supplementary information forming part of this submission.

**ORCID iD**
Piotr Ozieranski http://orcid.org/0000-0002-2023-3288

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
