## [Reviewer comments · BMJ Open]

ARTICLE DETAILS

TITLE (PROVISIONAL)	Accessibility and quality of drug company disclosures of payments to healthcare professionals and organisations in 37 countries: A European policy review
AUTHORS	Ozieranski, Piotr; Martinon, Luc; Jachiet, Pierre-Alain; Mulinari, Shai

VERSION 1 – REVIEW

REVIEWER	Lexchin, Joel York University, School of Health Policy & Management
REVIEW RETURNED	19-Jun-2021

GENERAL COMMENTS	This study examines the accessibility and quality of drug company payment data in European countries with different approaches to disclosure. This is important research as this type of information is key to understanding the depth of the relationship between the industry and healthcare practitioners and healthcare organizations. 1. The authors also need to recognize the limitations of transparency, e.g., that in the US it does not appear to lead to behavioural change – see: Lexchin et al. Journal of General Internal Medicine 2021: DOI: 10.1007/s11606-021-06657-0. 2. The authors also make the point about the payment data being
--

	available to the public is important. However, studies from the US show that the public does not access data in the Open Payments database. 3. For the countries where there was self-regulation the authors should indicate whether disclosure applied only to members of the industry association and trade groups or to all companies. 4. Page 4, lines 19-20: The authors should state what types of organizations and healthcare professionals are affected by the US Sunshine Act and which ones are not. 5. Page 4, line 29: It should be "prescribing" not "prescription". 6. Page 6, line 8: It should be "many fewer" not "much fewer". 7. Page 7, line 9: What do the authors mean by "generated observations"? 8. Page 10, line 7: What metric was used to measure "closeness"? Joel Lexchin Professor Emeritus York University Toronto, Canada
--	---

REVIEWER	Ozaki, Akihiko Minamisoma Municipal General Hospital, Department of Surgery
REVIEW RETURNED	28-Jul-2021

GENERAL COMMENTS	Many thanks for giving me a chance to review this article. Also, apologies for the delay in response. This is a well-designed research and will contribute to the ongoing debate on financial conflicts of interest between pharmaceutical companies and healthcare sectors. In further improving the manuscript, I have several comments as follows. Major comment 1. We applaud the authors' efforts to make comprehensive analyses for transparency initiatives adopted by European countries. However, because of the large amount of data, readers without
--

	enough knowledge may struggle to understand the flow of the argument, particularly, the results section. Minor comments. Abstract: 1. It was confusing that the sum of the countries with self-regulation (30) and countries with public regulations or a mix of self-regulation and public-regulation (15) exceeds the total number of 37 European countries. Please clarify.2. Page 2 Line 34-35, You may need to explain background information about EuroForDocs here, such as how they are integrating payment data. Without it, it is difficult to understand what figures such 56% and 100% mean.3. In conclusion, rather than just mentioning “transparency”, it may be better to refer to accessibility and data quality for consistency.4. In conclusion, “some improvements quality” should be “some improvements in quality”? I think this is a typo. Introduction 1. I was just wondering if it is better to use financial COIs than COIs given that you are not considering non-financial COIs? Further, in some parts, conflicts of interest are not abbreviated even after defining COI here. Methods 1. Page 6 Lines 50. It may be helpful if you can detail of 34 trade groups and 13 public/public-private bodies and their answers as in Supplementary Material? Results 1. With regards to section 3.1. Mapping regulatory approaches to payment disclosure, it is quite difficult to follow the flow of the argument at least for international readers without much knowledge on this field. This may be a natural consequence given that the authors include more than 30 countries in this study, but I suggest that the presentation can be more organized and sophisticated.
--	--

	2. I wondered how the authors ranked the self-regulation and public database in terms of data quality and accessibility, though I understand that the authors used specific criteria. It seems that the authors did not specifically calculate the score or the number of yes. Please specify. 3. Page 17. Please specify details of PDFs. Can you make "Copy and Paste"? Are characters identifiable? 4. Page 17. I would like to confirm that Irish data is searchable but not downloadable? I was a little bit confused. 5. About EuroForDocs, I just would like to confirm that they did not dare to extract image-based PDFs? Limitations 1. Page 26 Line 55. It may be good to detail that the EFPIA rule required the company to need consent from the individual HCPs in disclosing individual-level data.
--	--

REVIEWER	mitchell, aaron Memorial Sloan Kettering Cancer Center
REVIEW RETURNED	29-Jul-2021

GENERAL COMMENTS	This is an important and well-written investigation into the various systems in place for monitoring payments from the drug industry to physicians across European countries. While the results are not surprising, this study represents a large, thorough, multifaceted summary of the complexities of the current systems, providing a comprehensive scope of the regulatory environment. This is necessary work to lay the groundwork for future reform efforts, and the study authors present several recommendations for reform that stem directly from their observations in studying these various systems. I have several suggestions for improvement, mainly to help increase the clarity of the results presentation by giving more thorough definitions of the categories of regulation (self regulation vs. both
---

	kinds, etc.). Specific comments:  - I am not quite clear on the distinctions between the different categories (state regulation, self regulation, etc.). For example, for the term “countries pursuing self regulation” - does this mean, “countries that allow industry self-report of payment data but do not require it”? Or, “countries that require industry self-report of payment data”? Or something else? I understand that this term (and the other outcomes) is based on prior work which the authors cite, but it would be very helpful to briefly define each of these terms in the current introduction. Sections 2.2.1 and 2.2.3 mention the categories that were used, but do not go into detail regarding what each category means. - If the term “self regulation” does in fact mean something like “countries that require industry self-report of payment data”, then I wonder if there is better term to capture what is meant here. Simply because if the only requirement is to report what they are doing, and not actually change their behavior with respect to physician payments, then in some sense the industry payments are not really being “regulated.” And hence the term “self regulation” makes it sound like there is more action happening to change or decrease payments than is actually happening, which is in fact only disclosure. Would a term such as “industry self-report” be clearer in this regard? - It would be helpful early on to describe the EFPIA Code in more detail, so the readers can get a better sense of whether these “minimum requirements” are clearly inadequate, or whether they would actually do a good job if followed uniformly. - I am confused how 8 of the 30 countries with self-regulation also had public regulation, but then in the following paragraph (section 3.1) it is reported that only 4 countries had a combination of self and public regulation? I think these outcomes just need to be defined more clearly, so that it is clearer to the reader what the distinction between these 8 countries and 4 countries is. - The presented data in supplemental table 2 is not clear. Mainly, it
--	--

	isn't clear which of the groups of countries here correspond to which categories regulatory systems as discussed in the text – in other words, I can't get the numbers to match. For example, there are 30 countries described in the results section of the text as having self-regulation, but supplemental table 2 doesn't have a group with 30 countries, or multiple groups that would obviously be summed to get to 30. I think this is probably easily fixable with clearer labeling of the table, perhaps with color coding to delineate the sections, or something similar.  - Also on supplemental table 2, it is unclear to me why the bottom section (Belgium through Slovakia) has two types of regulation described next to it: both "single disclosure system..." and "two parallel systems..." - Are "information included" (table 2) and "characteristics included" (table 3) the same thing? - In Discussion, I would suggest a different term than "demonstrate" to describe the research results...perhaps "found" or "made findings which suggested that..."
--	---

VERSION 1 – AUTHOR RESPONSE

Responses to Reviewer #1

Thank you for committing your valuable time to reading and commenting on our manuscript. As always, we have benefitted significantly from your suggestions and list our detailed responses below.

	Reviewer's comments	Authors' responses (manuscript version with accepted changes)
--	---------------------	--

1	The authors also need to recognize the limitations of transparency, e.g., that in the US it does not appear to lead to behavioural change – see: Lexchin et al. Journal of General Internal Medicine 2021: DOI: 10.1007/s11606-021-06657-0.	Thank you for this excellent recommendation. We have now added a short paragraph at the end of the Discussion which indicates some of the key limitations of transparency and suggests how it should be complemented by other policy measures (see p. 31, last para).
2	The authors also make the point about the payment data being available to the public is important. However, studies from the US show that the public does not access data in the Open Payments database.	Thank you. We now refer to this issue in the new additional paragraph referred to above (see p. 31, last para).
3	For the countries where there was self-regulation the authors should indicate whether disclosure applied only to members of the industry association and trade groups or to all companies.	Thank you for suggesting this important clarification. We now make this point in the Discussion see p. 29, first para). We also have to note that establishing the extent to which companies which are not members of the national industry trade groups follow the codes of practice in relation to payment disclosure would require involve carrying out a separate project. Given that, first, this information is not provided systematically in the industry codes themselves and, second, the low response rate to of our email inquiries to the trade groups (only half of them responded), the only way to establish this would involve checking the list of companies disclosing payments against the list of members of pharma trade group in each of the 30 countries with self-regulation. The amount of work involved in this research would be significant as in some of these countries, such as Germany or Spain, there are well over 100 company members, and the checks would need to be conducted for two or three years to avoid false negatives.

4	Page 4, lines 19-20: The authors should state what types of organizations and healthcare professionals are affected by the US Sunshine Act and which ones are not.	Thank you for this comment. We made this point in the introduction, while comparing Open Payments to the European brand of industry self-regulation (p. 4, second para).
5	Page 4, line 29: It should be "prescribing" not "prescription".	Thank you. We have corrected this.
6	Page 6, line 8: It should be "many fewer" not "much fewer".	Thank you. We have corrected this mistake.
7	Page 7, line 9: What do the authors mean by "generated observations"?	We have clarified this point. See p. 7, first para.
8	Page 10, line 7: What metric was used to measure "closeness"?	We have now specified three levels of match between eurosfordocs.eu and the separately published industry summary data (p. 10, first para). These levels of match are necessarily arbitrary.

Responses to Reviewer #2

Thank you for reviewing our work so carefully and offering a range of very useful comments. We present our detailed responses in the table below. We appreciate your help in enhancing the quality of our work.

	Reviewer's comments	Authors' responses (manuscript version with accepted changes)
1	Major comment: 1. We applaud the authors' efforts to make comprehensive analyses for transparency initiatives adopted by European countries. However, because of the large amount of data, readers without enough knowledge may struggle to understand the flow of the argument, particularly, the results section.	Thank your positive appraisal of the manuscript. We appreciate this important comment. We agree entirely that it is key to ensure that readers with a general interest in this field can follow our argument. We sought to address your recommendation in the following ways.  • The overall structure of the findings is outlined in the first paragraph at the beginning of the Results section (p. 10). We have now added extra signposting and clarity in this paragraph. • We have now added more signposting throughout the Results to ensure greater coherence between its subsections and help readers follow connections between the bigger points we are making (p. 10, last para; p 22, first para in section 3.4).
2	Abstract: 1. It was confusing that the sum of the countries with self-regulation (30) and countries with public regulations or a mix of self-regulation- and public-regulation (15) exceeds the total number of 37 European countries. Please clarify.	Thank you for this helpful remark. We have now revised the sentence in question and added the numbers of countries. Because in 8 countries self-regulation is used in parallel with public regulation the total number of countries is higher than 37. We sought to make this point as clear possible within the word constraints available to the Abstract.

3	2. Page 2 Line 34-35, You may need to explain background information about EuroForDocs here, such as how they are integrating payment data. Without it, it is difficult to understand what figures such 56% and 100% mean.	We entirely agree with your point and have amended the abstract accordingly, making it clear what the two figures mean.
4	3. In conclusion, rather than just mentioning “transparency”, it may be better to refer to accessibility and data quality for consistency.	Thank you for this useful suggestion, we have implemented it.
5	4. In conclusion, “some improvements quality” should be “some improvements in quality”? I think this is a typo.	Thank you, we have eliminated this typo.
6	Introduction 1. I was just wondering if it is better to use financial COIs than COIs given that you are not considering non-financial COIs? Further, in some parts, conflicts of interest are not abbreviated even after defining COI here.	This is a very useful suggestion, thank you. We now use the “FCOI” abbreviation consistently throughout the article when referring to financial conflicts of interest.
7	Methods 1. Page 6 Lines 50. It may be helpful if you can detail of 34 trade groups and 13 public/public-private bodies and their answers as in	We have provided the background data on the responders and non-responders as a new Online supplement (Online Supplement 2).

	Supplementary Material?	We would not be able to include verbatim answers as we obtained them as emails and, within the context of European privacy laws, we would need to consent the senders again before their responses were made public. However, all the answers relevant for the purposes of this study are reported in the tables throughout the main body of the paper and in the Online Supplements. Of course, we can consider discussing answers received from specific organisations in more detail with interested readers who approach us once the paper is published.
8	Results 1. With regards to section 3.1. Mapping regulatory approaches to payment disclosure, it is quite difficult to follow the flow of the argument at least for international readers without much knowledge on this field. This may be a natural consequence given that the authors include more than 30 countries in this study, but I suggest that the presentation can be more organized and sophisticated.	Thank you for helping us improve this section. As you note, this section summarises information on 37 countries, including various combinations of different regulatory approaches. What's also important is that the interpretation of some of the country cases is interdependent. For example, the mention of a regulatory intervention in Greece is necessary for readers to understand the way we classify the Netherlands. All in all, we must present this detail to, first, avoid misrepresenting the data, second, to justify our interpretation of the more complex country cases, and, third, to allow readers to understand the argument made in the subsequent sections. We have addressed your helpful comment in the following ways to make our narrative as clear as possible for international readers.  • We have explained the meaning of self-regulation and government regulation more extensively in the introduction (p. 4, second para) • Our approach to the categorisation of regulatory approaches is now explained in much detail in the Methods (see the first four paragraphs in section 2.2.3). • We have added a Box (Box 1) on pp. 11-12 which is designed in a way that clearly shows the distribution of approaches to payment disclosure in different countries • We have revised section 3.1. significantly, with additional explanation of the differences

		between the regulatory approaches taken in different countries We would like to assure that we have tried various alternative ways of presenting the material in this section. The current write-up is the most straightforward given the number of countries and the level of complexity of the regulatory approaches.
9	2. I wondered how the authors ranked the self-regulation and public database in terms of data quality and accessibility, though I understand that the authors used specific criteria. It seems that the authors did not specifically calculate the score or the number of yes. Please specify.	We did not calculate scores because it would be difficult to reliably attribute weights to the selection criteria, which we note as a limitation of our study in the Discussion (see p. 29, last para). Instead, we used a heuristic, whereby the databases with higher overall data accessibility and quality were placed higher within the tables. This means that the databases with “more heat” would be placed towards the bottom of the tables. This is now explained in footnote 1 below Table 2 and footnote 1 below Table 3.
10	3. Page 17. Please specify details of PDFs. Can you make “Copy and Paste”? Are characters identifiable?	Thank you for this important question. We have now introduced further clarification into the text (p. 19, first para). More details of these PDF documents are provided in in another recent publication which is signposted in the methods section.¹
11	4. Page 17. I would like to confirm that Irish data is searchable but not downloadable? I was a little bit confused.	This was an inaccuracy, for which we apologise. Thank you for helping us spot it. One can only see the data by company, not search the name of a recipient. In addition, it is not possible to download the data. This has now been corrected in Table 2.

12	5. About EuroForDocs, I just would like to confirm that they did not dare to extract image-based PDFs?	Some of the image-based PDFs were extracted. Extensive details of how eurosfordocs.eu was created are provided in another recent publication which is signposted in the methods section (p. 7, first para). ¹
13	Limitations 1. Page 26 Line 55. It may be good to detail that the EFPIA rule required the company to need consent from the individual HCPs in disclosing individual-level data.	We now refer to this rule when introducing the concept of self-regulation in the Introduction (p. 4, second para).

Responses to Reviewer #3

Thank you for reviewing our work so carefully and offering many very useful recommendations. We present our detailed responses in the table below. We are grateful for your support in enhancing the clarity of our work.

	Reviewer's comments	Authors' responses and page numbers (manuscript version with accepted changes)
1	I am not quite clear on the distinctions between the different categories (state regulation, self regulation, etc.). For example, for the term “countries pursuing self regulation” - does this mean, “countries that allow industry self-report of payment data but do not require it”? Or, “countries that require industry self-report of payment data”? Or something else? I understand that this term (and the other outcomes) is based on prior work which the authors cite, but it would be very helpful to briefly define each of these terms in the current introduction. Sections 2.2.1 and 2.2.3 mention the categories that were used, but do not go into detail regarding what each category means.	Thank you for this helpful comment. We have now introduced an additional paragraph in the Introduction, which explains what is meant by self-regulation, government regulation and a combination of the two. These categories are then used systematically throughout the article (p. 4, second para) We also add further detail in section 2.2.1, where we explain the concept of “public regulation” (p. 7, third para), in section 2.2.3, where we explain our approach to the coding of different countries based on their regulatory approaches (p. 7, para 2-5), and throughout section 3.1, where we describe the distribution of the different regulatory approaches including their systematic descriptions (pp. 12-13). In responding to your specific questions regarding self-regulation, we now specify in section that it is the “default” approach in European countries whose pharmaceutical industry trade groups are EFPIA members (p. 13, first para).

		On top of this, we have ensured that all key concepts are now used consistently throughout the article. We have also reduced concepts that have been non-essential (such as “state”, “public-private”). We hope that this brings extra clarity to our argument.
2	If the term “self regulation” does in fact mean something like “countries that require industry self-report of payment data”, then I wonder if there is better term to capture what is meant here. Simply because if the only requirement is to report what they are doing, and not actually change their behavior with respect to physician payments, then in some sense the industry payments are not really being “regulated.” And hence the term “self regulation” makes it sound like there is more action happening to change or decrease payments than is actually happening, which is in fact only disclosure. Would a term such as “industry self-report” be clearer in this regard?	We appreciate your suggestion. Of course, the limited capacity of self-regulation to alter the behaviour of pharmaceutical companies in the field of payment disclosure is a key conclusion of our research. Building on your helpful comment, we now reinforce this point throughout the paper.  • In the Introduction we refer to examples of the limited effectiveness of self-regulation in relation to drug marketing in some European countries (p. 5, the para starting with “Second,...”). • In the Discussion we state clearly that disclosure does not equal transparency (p. 27, fourth para). • Also in the Discussion we now draw comparisons with other areas of public health policy where self-regulation has largely failed to modify corporate behaviour (p. 27, end of the fifth para). • Finally, in the concluding section, we point to broader limitations of transparency even in a well-functioning system like the US (p. 31, last para). Because the term “self-regulation” is used by the industry itself and in existing categorisations of approaches to payment disclosure, we would prefer to stick with it. Another reason to stick with this term is that because it is clearly identifiable it is easier to make

		policy points stressing the need to replace it with public regulation, as we do in the concluding section (p. 31, second para). Importantly, towards the end of the concluding section we now refer to recent research critiquing the extent of behavioural change achieved by the US-brand of public regulation of payment disclosure (p. 31, last para). This points to the limitations of transparency in managing COIs, whether it is achieved by public- or self-regulation. Even effective transparency (so not just disclosure) has its limits.
3	It would be helpful early on to describe the EFPIA Code in more detail, so the readers can get a better sense of whether these “minimum requirements” are clearly inadequate, or whether they would actually do a good job if followed uniformly	We provide a brief description of the key provisions of the EFPIA Code in the Introduction, while also comparing them with the US Sunshine Act (p. 4, second para). Further, in section 3.2 (and especially in the top part of Table 2) we summarise the minimum requirements specifically in relation to payment data accessibility and quality and contrast them with how they are followed by pharmaceutical industry trade groups in different countries. We also add a further point in the Discussion on the scope of companies following self-regulation which is directly relevant for the issue of non-disclosed payments.
4	I am confused how 8 of the 30 countries with self-regulation also had public regulation, but then in the following paragraph (section 3.1) it is reported that only 4 countries had a combination of self and public regulation? I think these outcomes just need to be defined more clearly, so that it is clearer to the reader what the distinction between these 8 countries and 4	Thank you for helping us clarify this point. In the methods we now explain our approach to classifying countries in more detail by stressing that it is possible for countries to two parallel approaches to disclosure (separate self- and public regulation) or have a single approach which combines elements of the two. With the same point in mind, we describe the distribution of the different regulatory approaches in section 3.1, while also summarising them in the new Box 1 (p. 11).

	countries is.	
5	The presented data in supplemental table 2 is not clear. Mainly, it isn't clear which of the groups of countries here correspond to which categories regulatory systems as discussed in the text – in other words, I can't get the numbers to match. For example, there are 30 countries described in the results section of the text as having self-regulation, but supplemental table 2 doesn't have a group with 30 countries, or multiple groups that would obviously be summed to get to 30. I think this is probably easily fixable with clearer labeling of the table, perhaps with color coding to delineate the sections, or something similar.	We do appreciate these helpful suggestions. We have now redesigned the table in question as Box 1.
	Also on supplemental table 2, it is unclear to me why the bottom section (Belgium through Slovakia) has two types of regulation described next to it: both "single disclosure system..." and "two parallel systems..."	As above, these issues have been fixed in the redesigned Box 1.
	Are "information included" (table 2) and "characteristics included" (table 3) the same	Thank you for spotting this inconsistency in labelling. We have now replaced "information" with "characteristics" in table 1, 2 and 3.

	thing?	
	In Discussion, I would suggest a different term than “demonstrate” to describe the research results...perhaps “found” or “made findings which suggested that...”	Thank you, we have replaced the word “demonstrate” with “suggest”.

References

- Mulinari S, Martinon L, Jachiet P-A, et al. Pharmaceutical industry self-regulation and non-transparency: country and company level analysis of payments to healthcare professionals in seven European countries. *Health Policy* 2021 doi: <https://doi.org/10.1016/j.healthpol.2021.04.015>

VERSION 2 – REVIEW

REVIEWER	Lexchin, Joel York University, School of Health Policy & Management
REVIEW RETURNED	04-Sep-2021

GENERAL COMMENTS	The changes made by the authors have dealt with my initial concerns. There is one final matter that I want to raise. Why in the face of inadequate regulation either by industry or government has little further action been taken to correct the limitations of the current reporting system. What are the reasons for the lack of initiatives at all levels – government, industry, healthcare professional associations and healthcare organizations. This article doesn't need to answer this question, but I think that it is important to raise the question as a topic of important and necessary research. Without answering this question, the suggestions by the authors for further reforms will likely go nowhere.
---

REVIEWER	Ozaki, Akihiko Minamisoma Municipal General Hospital, Department of Surgery
REVIEW RETURNED	04-Sep-2021

GENERAL COMMENTS	Many thanks for giving me an opportunity to review a revised version the manuscript. The authors properly revised the manuscript as per the requests from the editor and reviewers. I recommend the publication of this manuscript in its current form.
--

REVIEWER	mitchell, aaron Memorial Sloan Kettering Cancer Center
REVIEW RETURNED	07-Sep-2021

GENERAL COMMENTS	The authors have responded to all of my initial comments. The increased discussion of the definition of “self regulation” early on is very helpful, as is the description and distinction between parallel self-regulation and public-regulation vs. a combination of the two. Along with the new Box 1, this greatly increases the clarity of the outcome definition and the ease with which the reader can see which countries fall in to each category.
--

VERSION 2 – AUTHOR RESPONSE**Responses to Reviewer #1**

The changes made by the authors have dealt with my initial concerns

Thank you for your generous help in enhancing our manuscript. We are glad that you are satisfied with how we addressed your initial concerns.

There is one final matter that I want to raise. Why in the face of inadequate regulation either by industry or government has little further action been taken to correct the limitations of the current reporting system. What are the reasons for the lack of initiatives at all levels – government, industry, healthcare professional associations and healthcare organizations. This article doesn't need to answer this question, but I think that it is important to raise the question as a topic of important and necessary research. Without answering this question, the suggestions by the authors for further reforms will likely go

We do appreciate the very important questions that you have raised around the lack of momentum for improvement. This set of questions does indeed set agenda for future work. In responding to your query, we have made the following points.

First, we have reported available evidence on any corrective actions considered or undertaken by EFPIA and in countries with public regulation (new para towards the bottom of p. 23).

“EFPIA is clearly aware of at least some of the problems in payment data accessibility. For example, in 2019, it listed “improv[ing] access” via “[c]reateing platforms with [a] searchable tool” as one of the “main topics” to be considered by EFPIA itself and its member trade groups.¹ However, little evidence exists of subsequent discussions on this issue except for a planned “feasibility study” of possible “options for improving the disclosure” to be considered from 2021 to 2023.² Further, EFPIA does not seem to have recognised or engaged with the issues of low payment data quality.”

Second, your very useful comments have helped us make we an additional point about reporting challenges in countries with two parallel self- and public regulatory approaches (p. 24, end of second para)

“Problems with underreported payments may be particularly prominent in countries with parallel self- and public regulation due to possible confusion relating to where payments should be reported. For example, some healthcare organisations in England underreported some of the payments they had received given their implicit or explicit expectations that the payments would be disclosed via self-regulation.^{3 4}”

Third, we have also made a new point about available evidence on corrective actions in countries with public regulation (p. 24, last para)

“Inconsistencies in the approaches to public regulation between European countries are highlighted by EFPIA and used as a key argument in favour of self-regulation, which, in EFPIA’s words, represents a “global and consistent approach for companies across Europe and common understanding for the public”.¹ France is one country in which problems in data accessibility and “ergonomics” have been recognised by the Ministry of Health in 2018.⁵ Following this, a new version of *Transparence Santé* is due to be launched in late 2021, and is expected to adopt approaches to data presentation, including visualisations, similar to those developed earlier for *eurosfordocs.fr*. We are not aware of similar discussions in other countries with public regulation or combining self- and public regulation”

Finally, we have made the recommendation for future research on the limited extent of corrective action (p. 25, second para)

“Therefore, one key area of further study would involve using qualitative methods to identify and trace relationships between the likely causes of limited corrective action seeking to address the shortcomings of the current reporting systems in European countries. Of particular importance would be examining the incentive structures and motivations of public authorities, industry trade groups and companies, healthcare professional associations and patient organisations at the national and EU levels.”

Responses to Reviewer #2

Many thanks for giving me an opportunity to review a revised version the manuscript.

The authors properly revised the manuscript as per the requests from the editor and reviewers. I recommend the publication of this manuscript in its current form.

Thank you for this positive appraisal of our manuscript. We appreciate your valuable time in helping us improve this piece of work.

Responses to Reviewer #3

The authors have responded to all of my initial comments. The increased discussion of the definition of “self regulation” early on is very helpful, as is the description and distinction between parallel self-regulation and public-regulation vs. a combination of the two. Along with the new Box 1, this greatly increases the clarity of the outcome definition and the ease with which the reader can see which countries fall in to each category.

We are very pleased that the revised manuscript has addressed your initial concerns. Thank you very much for your excellent support in clarifying our argumentation.

Literature

1. EFPIA. Codes Committee Activities Report (2018) 2019 [Available from: <https://www.efpia.eu/media/554642/efpia-code-report-2018.pdf>].
2. EFPIA. EFPIA Report on Ethics & Compliance Activities (2020) 2021 [Available from: <https://www.efpia.eu/media/602865/efpia-code-report-2020-20210629.pdf>].
3. Moberly T. CCGs fail to declare pharma funding. *BMJ* 2018;360:j5911. doi: 10.1136/bmj.j5911
4. Moberly T. NHS joint working with industry is out of public sight. *BMJ* 2019;364:l1353. doi: 10.1136/bmj.l1353
5. Ministère des Solidarités et de la Santé. Discours d'Agnès Buzyn: remise du rapport sur l'amélioration de l'information des usagers et des professionnels de santé sur le médicament 2018 [updated 3rd September 2018. Available from: <https://solidarites-sante.gouv.fr/actualites/presse/discours/article/discours-d-agnes-buzyn-remise-du-rapport-sur-l-amelioration-de-l-information> accessed 29th October 2021.

VERSION 3 – REVIEW

REVIEWER	Lexchin, Joel York University, School of Health Policy & Management
REVIEW RETURNED	30-Oct-2021
GENERAL COMMENTS	The authors have dealt with my remaining suggestion.